# Watch and Wait Approach for Rectal Cancer Following Neoadjuvant Treatment: The Experience of a High Volume Cancer Center

**DOI:** 10.3390/diagnostics11081507

**Published:** 2021-08-21

**Authors:** Daniela Rega, Vincenza Granata, Carmela Romano, Valentina D’Angelo, Ugo Pace, Roberta Fusco, Carmela Cervone, Vincenzo Ravo, Fabiana Tatangelo, Antonio Avallone, Antonella Petrillo, Paolo Delrio

**Affiliations:** 1Colorectal Surgical Oncology, Department of Abdominal Oncology, Istituto Nazionale Tumori-IRCCS “Fondazione G. Pascale”, 80131 Naples, Italy; u.pace@istitutotumori.na.it (U.P.); carmela.cervone@istitutotumori.na.it (C.C.); p.delrio@istitutotumori.na.it (P.D.); 2Radiology Division, Istituto Nazionale Tumori-IRCCS “Fondazione G. Pascale”, 80131 Naples, Italy; v.granata@istitutotumori.na.it (V.G.); a.petrillo@istitutotumori.na.it (A.P.); 3Experimental Clinical Abdominal Oncology, Department of Abdominal Oncology, Istituto Nazionale Tumori-IRCCS “Fondazione G. Pascale”, 80131 Naples, Italy; c.romano@istitutotumori.na.it (C.R.); a.avallone@istitutotumori.na.it (A.A.); 4Gastroenterology and Endoscopy Unit, Department of Abdominal Oncology, Istituto Nazionale Tumori-IRCCS “Fondazione G. Pascale”, 80131 Naples, Italy; v.dangelo@istitutotumori.na.it; 5Medical Oncology Division, Igea SpA, 80100 Naples, Italy; r.fusco@igeamedical.com; 6Radiation Therapy, Istituto Nazionale Tumori-IRCCS “Fondazione G. Pascale”, 80131 Naples, Italy; v.ravo@istitutotumori.na.it; 7Pathology and Cytopathology Unit, Department of Support to Cancer Pathways Diagnostics Area, Istituto Nazionale Tumori-IRCCS “Fondazione G. Pascale”, 80131 Naples, Italy; f.tatangelo@istitutotumori.na.it

**Keywords:** rectal cancer, watch and wait, rectal sparing, no operative management, MRI

## Abstract

Multimodal treatments for rectal cancer, along with significant research on predictors to response to therapy, have led to more conservative surgical strategies. We describe our experience of the rectal sparing approach in rectal cancer patients with clinical complete response (cCR) after neoadjuvant treatment. We also specifically highlight our clinical and imaging criteria to select patients for the watch and wait strategy (w&w). Data came from 39 out of 670 patients treated for locally advanced rectal cancer between January 2016 until February 2020. The selection criteria were a clinical complete response after neoadjuvant chemotherapy managed with a watch and wait (w&w) strategy. A strict follow-up period was adopted in these selected patients and follow-ups were performed every three months during the first two years and every six months after that. The median follow-up time was 28 months. Six patients had a local recurrence (15.3%); all were salvageable by total mesorectal excision (TME). Five patients had a distant metastasis (12.8%). There was no local unsalvageable disease after w&w strategy. The rectal sparing approach in patients with clinical complete response after neoadjuvant treatment is the best possible treatment and is appropriate to analyze from this perspective. The watch and wait approach after neoadjuvant treatment for rectal cancer can be successfully explored after inflexible and strict patient selection.

## 1. Introduction

Total mesorectal excision (TME) is the current standard of care for mid and low locally advanced rectal cancers (LARC), defined as tumors staged T3 or above, or with lymph node involvement. The introduction of neoadjuvant chemo-radiotherapy (nCRT) has provided improvements in local disease control [1,2,3].

Although surgical techniques continue to improve, TME is correlated with a 1–2% rate of perioperative mortality, which increases with old age, frailty, and comorbidity [4]; it is also associated with a 31% rate of major complications (Clavien–Dindo grade 3–4) [5] such as anastomotic leaks, a 25% risk of a permanent stoma, chronic altered bowel function, or anorectal and sexual dysfunction in more than 60% of patients [6,7].

In approximately 15–27% of patients, neoadjuvant treatment may cause a complete pathologic response [8,9], which is associated with favorable long-term outcomes [10,11]. A multimodal diagnostic approach for optimal staging is therefore crucial in determining the appropriate strategy following neoadjuvant treatment. The role and accuracy of imaging in the detection of the primary tumor, residual rectal cancer or local recurrence appears crucial. For carefully selected patients with a significant tumor response at restaging, calculated as a decrease in tumor size as well as in the depth of tumor penetration in the wall, including lymph nodal sterilization, organ-preserving treatments could be explored such as a local excision and the w&w approach. These strategies aim at improving quality of life without compromising the oncological outcome.

The risk of local regrowth within 3 years after attaining a clinical complete response is 7–33% [12,13,14,15], and, therefore, long-term and more intensive follow-up strategies have been recommended for patients managed with a rectal preserving strategy [16,17].

We describe our experience of the rectal sparing approach in rectal cancer patients with clinical complete response (cCR) after neoadjuvant treatment. We also specifically underline our clinical and imaging criteria to select the patients for the watch and wait strategy.

## 2. Materials and Methods

### 2.1. Patient Population

Approval was obtained by the Institutional Review Board for this retrospective study and the requirement for informed consent was waived. We selected 128 patients with no signs of residual tumor at restaging exams and/or histopathologic exam (ycT0N0 or ypT0—TRG1 according to Mandard [18]) out of 670 patients treated with neoadjuvant therapy for locally advanced rectal adenocarcinoma from January 2016 to February 2020. Patient selection was performed through a computerized search in medical records. The inclusion criteria for the study population were the following:

(a)Patients with pathologically proven diagnosis of rectal cancer;(b)Patients who underwent neoadjuvant treatment such as only a short course of radiotherapy (25 Gy) or radio-chemotherapy (50 Gy with concomitant capecitabine at a daily dose of 825 mg/m^2^/12 h);(c)Patients who underwent digital rectal examination (DRE) and endoscopy within 15 days before treatment and who were re-evaluated after both DRE and endoscopy 8 weeks after treatment;(d)Patients who underwent magnetic resonance imaging (MRI) within 15 days before treatment and 8 weeks after the completion of neoadjuvant therapy;(e)Patients with no signs of residual tumor at restaging exams and/or histopathologic exam (ycT0N0 or ypT0—TRG1 according to Mandard [18]);(f)Patients who were followed for almost 1 year.

The exclusion criteria were the following:

(a)No accessible MRI and endoscopy study pre- or post-treatment;(b)Patients with residual tumor at restaging exams and/or histopathologic exam;(c)Unavailability for follow-up examination.

Out of this initial group, we considered a study group of 39 individuals and two control groups including 68 and 21 patients, belonging to the A and B control group, respectively. A scheme showing the composition of the three groups considered in this study is reported in Figure 1.

The final study population consisted of 39 patients (Table 1) identified as achieving a cCR, defined as no sign of residual tumor at restaging exams reported 8 weeks after the neoadjuvant treatment. These patients were managed by the w&w strategy. Out of this study group, 22 patients were also enrolled in a prospective multicentric trial aimed at investigating the role of both local excision and watch and wait approaches in patients treated with neoadjuvant therapy for rectal cancer, with major complete response (mCR) or cCR [19]. Out of the remaining 17 patients, we considered as selection criteria for the w&w strategy, beside those listed above, old age (more than 78 years of age), performance status (ECOG > 1), location of the tumor within 3 cm of the anal verge, and the refusal to undergo surgery.

In regards to the control groups, we identified:

Control Group A: 68 patients who underwent TME surgery from January 2016 to February 2020 with pathologic complete response at resection–ypT0N0—TRG1 (according to Mandard) [18].

Control Group B: 21 patients who underwent local excision (LE) from January 2016 to February 2020 with pathologic complete response in the specimen–ypT0Nx—TRG1 (according to Mandard) [18]. Three of these were also enrolled in a prospective multicentric trial, already described, investigating the role of both local excision and watch and wait approaches in patients treated with neoadjuvant therapy for rectal cancer, with mCR or cCR [19]. For the other 18 patients, the selection criteria, beside those given in detail later in the text, was local excision, old age, performance status (ECOG > 1), location of the tumor within 3 cm of the anal verge and the decision of the patient to preserve the rectum.

These three groups were not directly compared, given the retrospective design of the study, and there was absence of randomization to either TME or W&W or LE.

A specialized and dedicated multidisciplinary team (MDT) composed of colorectal surgeons, radiologists, medical oncologists, radiation oncologists, and pathologists attended regular meetings and discussed all relevant patients. For all patients we evaluated a physical examination including digital rectal examination (DRE), full blood count, liver and renal function tests, serum carcinoembryonic antigen (CEA), MRI to define local tumor status and computed tomography (CT) scan of thorax and abdomen to define presence of metastases. Positron emission tomography (PET) was required in selected patients to investigate suspected metastasis. For all patients, pre- and post-treatment data were assessed. Clinical and imaging procedures were carefully chosen following guidelines and existing evidence from previous studies [1,2,14,15,20].

All patients managed with rectal sparing (LE or w&w strategy) were specifically informed of the potential risks of these alternative treatments, of all benefits and of all the alternatives and were involved in the shared decision process.

### 2.2. Clinical Assessment

Evaluation of a clinical complete response (cCR) was based on the following criteria: absence of palpable mass at digital rectal examination; absence of residual tumor or ulceration or nodularity or stenosis in proctoscopy exam; possible white scar and teleangiectasias in proctoscopy exam; no residual tumor and no suspicious lymph nodes on MRI.

### 2.3. MR Imaging Protocol

According to our study protocol, an MRI examination was performed before and after nCRT with a 1.5 T scanner (Magnetom Symphony, Siemens Medical System, Erlangen, Germany) and phased-array body coil. Pre-contrast coronal T1-weighted two-dimensional (2D) turbo spin-echo (TSE) images, and sagittal and axial T2-weighted (T2w) 2D TSE images were acquired. Axial DW images were obtained with spin-echo diffusion- weighted echo-planar sequence (SE-DW-EPI) at b values equal to 0, 50, 100, 150, 300, 600, 800 s/mm^2^. Axial dynamic contrast-enhanced T1-weighted fast low angle shot three-dimensional gradient-echo images were obtained: 1 sequence before and 10 sequences after intravenous injection of 0.1 mmol/kg of a positive, gadolinium-based paramagnetic contrast agent (Gd-DOTA, Dotarem, Guerbet, Roissy-CdGCedex, France) at 2 mL/s of flow rate, followed by a 10 mL saline flush at the same rate. Sagittal, axial and coronal post contrast T1-weighted 2D TSE images, with and without fat saturation, were also acquired. Table 2 reports MR sequence parameters.

In order to reduce bowel spastic artefacts, each patient received bowel preparation and antispasmodic medication.

### 2.4. Image Analysis

Two expert radiologists assessed MRI studies according to the structured reporting of rectal cancer staging and restaging by the Italian Society of Medical and Interventional Radiology (SIRM) [21].

We assessed: residual tumor, MRI tumor regression grade (TRG) according to Dworak [22,23], restricted diffusion appearance, mucin response, ycT-stage, distance from the inferior border of the tumor to the anal verge, distance from the inferior border of the tumor to the anorectal junction, cranio-caudal tumor length, anal sphincter complex involvement, CRM involvement, relationship with anterior peritoneal reflection, lymph node metastases, tumor deposits and extramural vascular invasion (EMVI).

Residual tumor was defined as:No fully normalized rectal wall (complete response);No fibrotic thickening of the wall without a residual mass (complete or near full response);Residual mass.

Restricted Diffusion (RD) appearance was defined as

-Yes (RD);-No (RD).

Mucin response was defined as:-Mucin (or colloid degeneration) response in non-mucinous tumor;-Mucinous tumor without response.

Follow-up protocol for the patients managed with rectal sparing (both local excision and watch and wait approaches) included clinical evaluation, DRE, proctoscopy and CEA every three months (for the first two years and every six months for next three years), pelvis MRI every six months, contrast-enhanced whole-body CT every 8 months, colonoscopy after the first year and consecutively depending on clinic evaluation.

Follow-up protocol for the patients managed with TME included clinical evaluation and CEA every 4 months for the first two years and then every 6 months for next three years, imaging of both the chest and liver every year, MRI every year, colonoscopy after the first year and thereafter depending on clinical evaluation.

Local regrowth was defined as tumor regrowth in the rectal lumen or in regional lymph nodes, as identified by clinical evaluation, endoscopy or imaging. Local regrowth was an indication for salvage surgery via TME.

Distant metastases were defined as the presence of metastatic disease, as identified by radiological exam or confirmed histologically.

## 3. Statistical Analysis

Chi-squared test with Yates’s correction was employed to analyze differences in percentage values of categorical variables. A *p* value < 0.05 was considered as statistically significant. Statistical analysis was obtained using the Statistic Toolbox of Matlab (The MathWorks, Inc., Natick, MA, USA).

## 4. Results

### 4.1. Study Population

This cohort study was homogeneous for sex, age, high grade of tumor, TNM stage and neoadjuvant treatment performed.

Clinical examination showed:-No palpable nodule on DRE in all patients;-No stenosis following proctoscopy in all patients;-A completely normal mucosa after proctoscopy in 16 patients;-A little (less than 2 cm of diameter) flat white scar at proctoscopy in 19 patients;-Teleangiectasias at the time of proctoscopy in 4 patients.

Radiological analysis showed a complete response in 10 patients and in 29 a fibrotic response. No one had a mucinous response. Among patients with a complete response, we found no restricted diffusion; the restricted diffusion was found in two patients with fibrotic response. No patient showed EMVI or tumor deposits; in six patients we found residual (median of two per patient (range 1–3)) in mesorectal side, all < 5 mm (Table 3).

According to TNM classification, 8th Edition, AJCC-UICC 2017 [24], all patients were classified as Stage 0.

During follow-up (median 28 months; range 12–50 months) clinical examination and MRI, we found local disease recurrence in six patients (15.3%), after, respectively, 30 months, 30 months, 31 months, 10 months, 12 months and 12 months (mean 20.8 months, range 10–31 months).

Salvage surgery after TME with a curative intention was performed in four patients, and two patients refused it.

Pathologic exam following TME were: ypT2N0 TRG3, ypTisN0 TRG1, ypT2N0 TRG4, ypT1N0 TRG1 (Table 4). The next follow-up (median 16 months, range 5–45 months) did not show recurrence.

Five patients (12.8%) developed distant metastases, which were localized in the liver. Two patients were treated locoregionally, one was treated surgically, and two with systemic chemotherapy. None of these patients developed local recurrence.

### 4.2. Control Group A

In these 68 patients, clinical examination showed an incomplete clinical response after neoadjuvant treatment after both DRE and proctoscopy, with a palpable mass of more than 2 cm in diameter, and/or fixed lesion on the rectal wall.

Radiological analysis showed remaining tumor in 54 patients with a fibrotic response in 14 patients. Five patients showed a mucinous response. We found restricted diffusion in all patients. All patients exhibited residual lymph nodes (median 4 per patients (range 3–6) in mesorectal side, all nodes were ≥5 mm). In ten patients we found tumor deposits (median 2 per patient (range 1–3)) with fibrotic response. In three patients we found residual extramural venous invasion (EMVI) (Table 3).

According to TNM classification, 8th Edition, AJCC-UICC 2017 [24], all patients were classified as Stage 1 or more.

During follow-up (median 28 months; range 12–50 months) we found local recurrence in two patients (2.9%); no patient developed distal metastasis.

### 4.3. Control Group B

In these 21 patients, clinical examination showed a major response to neoadjuvant treatment, with small superficial soft irregularity after DRE and small mucosal irregularity no more than 2 cm in diameter at endoscopy.

Radiological analysis showed that in 14 patients we found remaining tumor with a fibrotic response; in 7 patients we found fibrotic response. No patients showed mucinous response. We found restricted diffusion in 20 patients, all with residual tumor. All patients showed residual nodes (median 2 for patients (range 1–4) in mesorectal side, all nodes were <5 mm). No patients had tumor deposits or EMVI (Table 3).

According to TNM classification, 8th Edition, AJCC-UICC 2017 [24], all patients were classified as Stage 0 or 1.

During follow-up (median 28 months; range 12–50 months), we found disease recurrence in two patients (9.5%), after 10 and 13 months, respectively.

All patients were treated by salvage surgery with total mesorectal excision (TME) with curative intention; at pathological exam the local recurrence was only in the rectal wall, not in regional nodes. No patient developed distant metastasis. The next follow-up (median 5 months, range 1–27 months) did not show recurrence.

### 4.4. Statistical Results

There were statistically significant differences in radiological response rate among groups (*p* value << 0.001 at chi-squared test, Table 3). The statistically significant radiological findings among the three groups were presence of residual mass or fibrotic thickening of the wall, restricted diffusion, and residual lymph nodes presence and size.

There was no difference statistically significant in recurrence rate among groups (*p* value = 0.13 at chi-squared test, Table 3).

## 5. Discussion

The watch and wait strategy is being used more and more as a treatment option for patients with clinical complete response after neoadjuvant treatment for rectal cancer. The diagnosis of a cCR based on the results of clinical exam (DRE, proctoscopy and MR) is not always perfectly related to a real pathologic response. Local regrowth rates within 2 years are described in a range from 7 to 33% [12,13,14,15].

The risk of local regrowth or distant metastases after non-operative management in patients with cCR to neoadjuvant treatment remains a challenge for clinicians that conduct the patient care pathway.

In this study we explored our current clinical strategy to predict complete clinical response to neoadjuvant treatment for rectal cancer, reporting two groups of patients with pathologic complete response and a group with clinically complete response, treated during the same period.

The MDT is guided by clinical and MRI assessment to predict the pathological response to neoadjuvant treatment for rectal cancer [25,26,27], and a non-operative management approach may be considered only in centers with experienced multidisciplinary teams.

Magnetic resonance imaging (MRI) is the most accurate test to define locoregional clinical staging. By discovering extramural vascular invasion (EMVI) [28] and determining the T substage and distance to the circumferential resection margin (CRM), MRI can also predict the risks of local recurrence and synchronous/metachronous distant metastases, and should be carried out to select patients for their respective preoperative management and to define the extent of surgery. A standard proforma for MRI and pathology ensures a comprehensive report. The version of TNM staging used by the histopathologist and the MDT should be documented, acknowledged by all members of the MDT and regularly updated.

In our study we reported that there was statistically significant difference in radiological response rate among groups: the radiological findings were statistically significant among the three groups and included presence of residual mass or fibrotic thickening of the wall, restricted diffusion, and residual lymph nodes presence and size.

In our experience, the mean time to the local regrowth during follow-up was 20.8 months (range 10–31 months), with a higher rate of local recurrences within three years compared to surgical groups (15.3% vs. 2.9% and 9.5%). This seems to be lower than results from other authors who reported a larger data series of w&w strategy [12,29,30,31], and higher than a multicentric registry study [32] with a very large follow-up.

Local regrowth was diagnosed in three patients in the first few years after the end of neoadjuvant treatment; for another three patients, local regrowth occurred within the first three years from the end of treatment; all tumors were located within the bowel wall, and all were salvageable.

The local recurrence rate after salvage surgery in the w&w strategy group was comparable to that of the surgical group, indicating that delayed surgery may not compromise the local control in these selected patients. Rectal preservation was achieved in 32 out of 39 patients (82%) in the w&w group.

In our study we reported that there was no statistically significant difference in the recurrence rate among groups.

Even though, in our series, distant metastases only appeared in the group of w&w strategy, it is difficult to establish if there was a causal relationship with the whole organ preservation, due to the small size of the series. However, the rates were lower than those with similar pretreatment tumor stages [33,34], indicating that a cCR could have less aggressive biological behavior and is associated with favorable prognoses.

No patients with distant metastases had local regrowth, supporting the feasibility of the w&w option, and the low impact on distant sites. This, in contrast with other descriptions [12,15], shows a higher distant metastasis rate in patients with local regrowth compared to those who did not have local regrowth.

Our data suggest that patients with an accurate evaluation of cCR and those treated with the w&w strategy have no oncological disadvantage, and outcome seems to be comparable to those with complete pathologic response after TME surgery or LE surgery. Selection and surveillance of these patients should be performed in dedicated centers.

Our findings support the hypothesis that, in selected patients with rectal cancer, organ preservation after neoadjuvant treatment can be achieved avoiding an aggressive surgical approach with related post-operative complications and functional disorders due to TME.

## 6. Conclusions

The key to proposing a w&w strategy with no significant compromises in oncological outcomes is the inflexible selection of the patient, using three simple and clear parameters: no palpable mass to DRE, no residual tumor to proctoscopy, no residual tumor or nodes to MRI. Consecutively, an intensive surveillance is mandatory, to allow a timely surgical control in case of local regrowth.

The limitations of this study are the relatively short follow-up and long-term survival data.

## Figures and Tables

**Figure 1 diagnostics-11-01507-f001:**
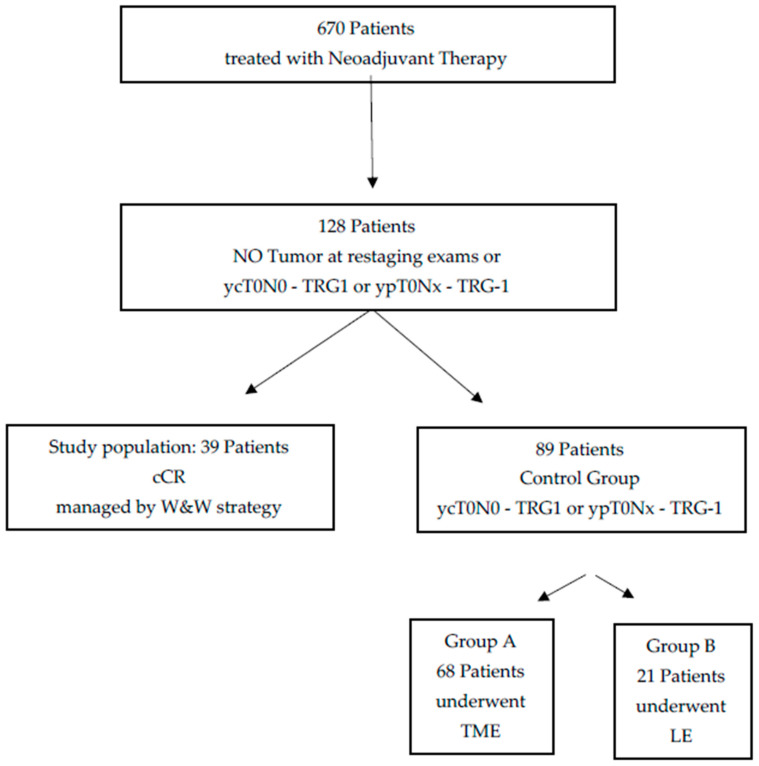
Scheme of groups compositions in this study. The following acronyms are used: clinical complete response (cCR), total mesorectal excision (TME), watch and wait (W&W), local excision (LE).

**Table 1 diagnostics-11-01507-t001:** Clinical and demographic characteristics of the W&W group.

Age, Mean (Range), Years	66, (46–84 aa)
Sex	
Men	29
Women	10
Height from anal verge, mean (range), cm	6 cm (1–12 cm)
Clinical tumor (T) classification	
T2	12
T3	25
T4	2
Clinical nodal (N) classification	
N0	7
N+ positive	32
Neoadjuvant regimen	
50 Gy with Capecitabine	28
25 Gy	11

**Table 2 diagnostics-11-01507-t002:** MR acquisition protocol.

Sequence	Orientation	TR/TE/FA (ms/ms/degree)	FOV (mm × mm)	Pixel Spacing	ST/Gap (mm/mm)
T1w 2D TSE	Coronal	499/13/150	450 × 450	0.87 × 0.87	3/0
T2w 2D TSE	Sagittal	4820/98/150	250 × 250	0.78 × 078	3/0
T2w 2D TSE	Axial	3970/98/150	250 × 250	0.78 × 0.78	3/0
SE-DW-EPI	Axial	2700/83	270 × 230	1.70 × 1.70	4/0
T1w FLASH 3D	Axial	9.8/4.76/25	330 × 247	0.59 × 0.59	3/0
T1w FLASH 3D	Axial	9.8/4.76/25	330 × 247	0.59 × 0.59	3/0
T1w 2D TSE	Sagittal	538/13/150	250 × 250	0.48 × 0.48	3/0
T1w 2D TSE	Coronal	538/13/150	250 × 250	0.48 × 0.48	3/0
T1w 2D TSE	Axial	450/12/150	270 × 236	0.52 × 0.52	3/0

AT, acquisition time; DW, diffusion weighted; EPI, echo-planar sequence image; FA, flip angle; FLASH, fast low angle shot; FOV, field of view; SE, spin echo; ST, slice thickness; TE, echo time; TF, turbo factor; T1w, T1-weighted; TSE, turbo spin echo; T2w, T2-weighted; TR, repetition time; 2D, two dimensional; 3D, three dimensional.

**Table 3 diagnostics-11-01507-t003:** Radiological findings in the three patient groups (Study population, Control Group A, Control Group B).

	Radiological Findings							
	Radiological Response			Restricted Diffusion	EMVI Presence	Residual Lymph Nodes		Recurrence Rate
	Complete Response	Fibrotic Thickening of the Wall without a Residual Mass	Residual Mass			Diameter < 5 mm	Diameter ≥ 5 mm	
Study population N. 39)	10/39	29/39	0/39	0/39	0/39	6/39	0/39	6/39
Control Group A (N. 68)	0/68	14/68	54/68	68/68	3/39	0/68	68/68	2/68
Control Group B (N. 21)	0/21	7/21	14/21	20/21	0/21	21/21	0/21	2/21
*p*-value at chi-squared test	<0.001			<0.001	0.42	<0.001		0.13

**Table 4 diagnostics-11-01507-t004:** Clinical characteristics of patients with local regrowth and subsequent salvage surgery.

Patient	Distance from Anal Verge, cm	Initial Clinical Staging	pCRM	Neoadjuvant Treatment	Time to Regrowth	Pattern of Regrowth	Salvage Surgery	Pathology Staging	pCRM	Distant Metastases
1	1	T2N1	Negative	50 Gy + capecitabine	30 months	Endoluminal	LAR	ypT2N0–TRG 3	Negative	No
2	2	T3N1	Negative	50 Gy + capecitabine	30 months	Endoluminal	LAR	ypTisN0–TRG 1	Negative	No
3	8	T3N2	Negative	50 Gy + capecitabine	31 months	Endoluminal	LAR	ypT2N0–TRG4	Negative	No
4	6	T3N1	Negative	50 Gy + capecitabine	10 months	Endoluminal	APR	ypT1N0–TRG2	Negative	No
5	10	T3N1	Negative	25 Gy	12 months	Endoluminal	Refused			No
6	12	T4N2	Negative	50 Gy + capecitabine	12 months	Endoluminal	Refused			No

## Data Availability

The data presented in this study are available on request from the corresponding author.

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
