# Peer review of "Watch and Wait Approach for Rectal Cancer Following Neoadjuvant Treatment: The Experience of a High Volume Cancer Center"

_diagnostics, 2021, doi:10.3390/diagnostics11081507_

Round 1

Reviewer 1 Report

The authors specify that they included 128 patients in the study, but in the next paragraph they specify that there were only 39. After that the authors speak about 3 groups but they describe just 2. Could the authors please explain better.

Please rewrite the bibliography because articles 5, 6, 8, 9,11.

Please also cite more recent articles.

Author Response

We thank the Editor and the Reviewer for their interesting observations and constructive comments. In the following, we report our answers (in red) to specific comments. On their basis we have amended the manuscript using changes track. We are confident that now the manuscript will encounter their satisfaction.

Point 1. The authors specify that they included 128 patients in the study, but in the next paragraph they specify that there were only 39. After that the authors speak about 3 groups but they describe just 2. Could the authors please explain better.

Response 1. We thank the reviewer for giving us the opportunity to clarify this important point. We selected 128 patients with no sign of residual tumor out of 670 patients treated with neoadjuvant therapy for locally advanced rectal adenocarcinoma (from January 2016 until February 2020). The inclusion criteria for the selection were described in the 'materials and methods' paragraph (new lines from 79 to 90). This group was then divided in two control groups and a study population including 39 patients who were identified as achieving a cCR after the neoadjuvant treatment, and managed by the wait-and-watch strategy.

More precisely the two control groups were characterized as in the following:

  • Control Group A: 68 patients who underwent TME surgery with pathologic complete response at resection – ypT0N0 - TRG1 (according to Mandard);
  • Control Group B: 21 patients who underwent local excision (LE) with pathologic complete response in the specimen – ypT0Nx - TRG1 (according to Mandard). 

In order to be more clear about the composition of the study and control groups we included a scheme in the revised manuscript (figure 1). 

Point 2. Please rewrite the bibliography because articles 5, 6, 8, 9,11.

Response 2. We revised the references of articles 5, 6, 8, 9, 11, as required.

Point 3. Please also cite more recent articles.

Response 3. We added some recent articles, as required. In particular, we added new reference 17, including a study on radiation treatment in neoadjuvant for rectal cancer; new references 23, 29, 30, 31 looking at validation of the standardized index to analyze DCE-MRI data in the assessment of neo-adjuvant therapy in rectal cancer and to predict pathological complete response, and MRI detection of extramural venous invasion in rectal cancer.

Reviewer 2 Report

MAJOR POINTS

  • A systematic comparison of the study findings with those from the existing literature is lacking and should be performed. Also the clinical and imaging criteria for patient selection should be compared to those from the existing literature.
  • It is not clear what advancements to knowledge are brought by the study findings - this point should be emphasised to highlight how and to what extent they could be useful for patient management and applied to clinical practice.
  • The English language of the manuscript and its overall style need to be substantially improved.

SPECIFIC COMMENTS

  • Patient inclusion criteria should be listed in a separate table for better clarity.
  • Lines 137-138. Please specify the molecule of the contrast medium used for dynamic contrast-enhanced imaging. Furthermore, why was contrast medium used in this setting, given that contrast medium administration is usually considered unnecessary for rectal cancer staging and post-CRT restaging?
  • Lines 140-141. The sentence: "Table 2 reports MR sequence parameters" should be modified into: "Table 3 reports...", as Table 2 lists the "clinical characteristics of patients with local regrowth and subsequent salvage surgery".
  • Lines 170 and 174. What do you mean by "imaging of the chest and liver every 8 months"? Contrast-enhanced CT? Please specify.
  • Table 4 (lines 203-204). Please replace "p<<0.001" with "p<0.001".

Author Response

We thank the Editor and the Reviewer for their interesting observations and constructive comments. In the following, we report our answers (in red) to specific comments. On their basis we have amended the manuscript using changes track. We are confident that now the manuscript will encounter their satisfaction.

MAJOR POINTS

  • A systematic comparison of the study findings with those from the existing literature is lacking and should be performed. Also the clinical and imaging criteria for patient selection should be compared to those from the existing literature.

Response. We thank the Reviewer for giving us the opportunity to improve the discussion regarding the novelty (see also our response to the following point) of the study and the comparison with existing literature. The w&w strategy is a relatively recent and uncommon approach, and in the Discussion Section (new page 9, new lines 612-614) we considered sufficient the comparison of our study with previous results from our Institution, with authors who already reported a large data series of w&w strategy  (refs 25, 32, 33, 34)  and with a multicentric registry study (ref. 35). Moreover, clinical and imaging procedures were carefully chosen following guidelines and existing evidence from previous studies (ref. 1, 2, 14, 15, 20). We included a paragraph in the Methods Section, Population subsection, to clarify the rationale of our choices.

  • It is not clear what advancements to knowledge are brought by the study findings - this point should be emphasised to highlight how and to what extent they could be useful for patient management and applied to clinical practice.

Response. We thank the Reviewer, we expanded the final part of the Discussion to better emphasize and highlight the importance in the community of our results. In particular in patients with rectal cancer, organ preservation after neoadjuvant treatment can be achieved in selected cases avoiding an aggressive surgical approach with related post-operative complications and functional disorders related to rectal anterior resection.

  • The English language of the manuscript and its overall style need to be substantially improved.

Response: English language of the manuscript has been carefully revised by a native language.

SPECIFIC COMMENTS

  • Patient inclusion criteria should be listed in a separate table for better clarity.

Response. Patient inclusion criteria and organization of the three groups (study, control A and control B groups) have been deeply revised in the manuscript, to encounter requests of the Reviewer 1 too. A scheme (Figure 1) has been also introduced to facilitate the illustration.

Point . Lines 137-138. Please specify the molecule of the contrast medium used for dynamic contrast-enhanced imaging. Furthermore, why was contrast medium used in this setting, given that contrast medium administration is usually considered unnecessary for rectal cancer staging and post-CRT restaging?

Response . We clarified as suggested modifying the sentence as follow: Axial dynamic contrast-enhanced T1-weighted fast low angle shot three-dimensional gradient-echo images were obtained: 1 sequence before and 10 sequences, after intravenous injection of 0.1mmol/kg of a positive, gadolinium-based paramagnetic contrast agent (Gd-DOTA, Dotarem, Guerbet, Roissy-CdG; Cedex, France) at 2ml/s of flow rate, followed by a 10 ml saline flush at the same rate. Sagittal, axial and coronal postcontrast T1-weighted 2D TSE images, with and without fat saturation were also acquired.

We corrected the sentence, introduced according our study protocol; although the contrast medium is not necessary in the diagnosis of rectal cancer, it is shown in the literature that the sequences in DCE-MRI improve the diagnostic accuracy in the differential diagnosis between tumor residue and fibrosis.

Point . Lines 140-141. The sentence: "Table 2 reports MR sequence parameters" should be modified into: "Table 3 reports...", as Table 2 lists the "clinical characteristics of patients with local regrowth and subsequent salvage surgery".

Response . We corrected as suggested.

Point . Lines 170 and 174. What do you mean by "imaging of the chest and liver every 8 months"? Contrast-enhanced CT? Please specify.

Response. We modified the sentence as 'Contrast-enhanced CT of the chest and liver...'

Point. Table 4 (lines 203-204). Please replace "p<<0.001" with "p<0.001"

Response. We corrected as suggested.

Round 2

Reviewer 2 Report

Some points need to be improved:

-Line 722-729 "Our data supports that patients with an accurate evaluation of cCR have no oncological disadvantage from those with the w&w strategy, and seems to be comparable to those with complete pathologic response after TME surgery or LE surgery. Selection and surveillance of these patients should be performed in dedicated centers.

Our findings support the hypothesis that, in patients with rectal cancer, organ preservation after neoadjuvant treatment can be achieved in selected cases avoiding an aggressive surgical approach with related post-operative complications and functional disorders related to rectal anterior resection." It's not clear. Please rephrase it

- Line 453. "Contrast-enhanced CT of the chest and liver" it's not clear. I think it would be better "Contrast-enhanced whole-body CT". Lines 452 and 457. What do you mean by "MRI every six months"? Rectal MRI? Please specify.

- Line 554 "onesurgerically" Line 576 "ramaining tumor" and other.. The English language of the manuscript and its overall style need some minor check

Author Response

We thank the Editor and the Reviewer for their constructive observations. In the following, we report our answers (in red) to specific comments. On their basis we have amended the manuscript using changes track. We are confident that now the manuscript will encounter their satisfaction.

-Line 722-729 "Our data supports that patients with an accurate evaluation of cCR have no oncological disadvantage from those with the w&w strategy, and seems to be comparable to those with complete pathologic response after TME surgery or LE surgery. Selection and surveillance of these patients should be performed in dedicated centers.

Our findings support the hypothesis that, in patients with rectal cancer, organ preservation after neoadjuvant treatment can be achieved in selected cases avoiding an aggressive surgical approach with related post-operative complications and functional disorders related to rectal anterior resection." It's not clear. Please rephrase it

We thank the reviewer, we rephrased the paragraph in the following way and hope we clarified the meaning:

Changes in the manuscript: “Our data support that patients with an accurate evaluation of cCR and treated with the w&w strategy have no oncological disadvantage, and outcome seems to be comparable to those with complete pathologic response after TME surgery or LE surgery. Selection and surveillance of these patients should be performed in dedicated centers.

Our findings support the hypothesis that, in selected patients with rectal cancer, organ preservation after neoadjuvant treatment can be achieved avoiding an aggressive surgical approach with related post-operative complications and functional disorders due to TME."

- Line 453. "Contrast-enhanced CT of the chest and liver" it's not clear. I think it would be better "Contrast-enhanced whole-body CT". Lines 452 and 457. What do you mean by "MRI every six months"? Rectal MRI? Please specify.

We thank the reviewer, we substituted as suggested

Contrast-enhanced CT of the chest and liver  Contrast-enhanced whole-body CT

And clarified

MRI every six months  Pelvis MRI every six months

- Line 554 "onesurgerically" Line 576 "ramaining tumor" and other.. The English language of the manuscript and its overall style need some minor check

We thank again the reviewer. We corrected

Onesurgerically  one was treated surgically

And ramaining tumor  residual tumor

We also checked typos and style

This manuscript is a resubmission of an earlier submission. The following is a list of the peer review reports and author responses from that submission.